# Gaussian Process Predictions with Uncertain Inputs Enabled by Uncertainty-Tracking Processor Architectures

**Janith Petangoda**[*]
Signaloid,
Cambridge, UK

**Chatura Samarakoon**
University of Cambridge,
Cambridge, UK

**Phillip Stanley-Marbell**
Signaloid / University of Cambridge,
Cambridge UK.

## Abstract

Gaussian Processes (GPs) are theoretically-grounded models that capture both aleatoric and epistemic uncertainty, but, the well-known solutions of the GP predictive posterior distribution apply only for deterministic inputs. If the input is uncertain, closed-form solutions aren't generally available and approximation schemes such as moment-matching and Monte Carlo simulation must be used. Moment-matching is only available under restricted conditions on the input distribution and the GP prior and will miss the nuances of the predictive posterior distribution; Monte Carlo simulation can be computationally expensive. In this article, we present a *general* method that uses a recently-developed processor architecture [1, 2] capable of performing arithmetic on distributions to implicitly calculate the predictive posterior distribution with uncertain inputs. We show that our method implemented to run on a commercially-available implementation [3] of an uncertainty-tracking processor architecture captures the nuances of the predictive posterior distribution while being $\sim$108.80x faster than Monte Carlo simulation.

## 1 Introduction

Gaussian Processes (GPs) are a class of non-parametric models that can intrinsically capture model uncertainty. The prediction of a GP for a unobserved input is referred to as the *predictive posterior distribution* since it is a distribution over the predictive output conditioned on the observed data. For deterministic inputs, there is a closed-form solution for the predictive posterior distribution of a GP.

GPs can also capture the impact of *input uncertainty* on the predicted output. With an uncertain input, the GP predictive posterior distribution is in general arbitrary and no general closed-form solution exists. A typical tactic for handling uncertain inputs is to carry out moment-matching, where the first $k$ (usually $k = 2$) moments of the true predictive posterior distribution are calculated [4, 5]. Closed-form solutions for even the first two moments are only available in special cases, and even then, such a Gaussian approximation loses key statistical properties of the true posterior distribution, such as multi-modality and skewness. A better approximation could be obtained by carrying out a (potentially) computationally-expensive Monte Carlo simulation.

In this article, we present an algorithm for implicitly computing the GP predictive posterior distribution with uncertain inputs using recent advances in uncertainty-tracking processor architectures (UTPAs) [1, 2]. Our method does not rely on Monte Carlo simulation but rather exploits the internal distributional representation of the UTPA. Our method is *agnostic to the distribution of the input and the GP prior*.

We evaluate the effectiveness of our method against the Monte Carlo method by comparing the accuracies of the solutions to the ground-truth (obtained by a Monte Carlo simulation with a large

---

[*]The work was mostly carried out at the University of Cambridge.

Second Workshop on Machine Learning with New Compute Paradigms at NeurIPS 2024 (MLNCP 2024).

number of samples). We also compare the run time of each method. We show in Section 6 that running our new algorithm on a commercially-available implementation [3] of a UTPA [1, 2] can achieve orders of magnitude faster performance while achieving the same accuracy.

## 2 Gaussian Processes

A GP is a collection of random variables (Definition A.2 in Appendix A) where any *finite* subcollection of those random variables is jointly Gaussian. It is a distribution over the *infinitely*-many random variables that satisfy this condition on any *finite* subcollection of them. We can define a GP implicitly by specifying a mechanism for deriving the mean vector and covariance matrix of the multivariate Gaussian distribution that distributes any given subset of random variables. We can do so using a mean function $m$ and covariance kernel $k$ to derive the mean vector and covariance matrix of the multivariate Gaussian distribution using the *indexes* of the random variables.

**Definition 2.1** (Implicit definition of a Gaussian Process)**.** Let $X$ be an index set. Then, for a finite subset of $X$, $\{x_1, ..., x_n\} \in X$, let the random variables $Y_{x_1}, ..., Y_{x_n}$ be jointly distributed as a multivariate Gaussian, $\mathcal{N}(\mu, \mathbf{\Sigma})$, where the elements $\mu_i$ of the mean vector $\mu$ and $\Sigma_{ij}$ of the covariance matrix $\mathbf{\Sigma}$ are implicitly derived using

$$\mu_i = m(x_i),$$

and

$$\mathbf{\Sigma}_{ij} = k(x_i, x_j).$$

The function $m : X \to \mathbb{R}$ is called the *mean function* and the function $k : X \times X \to \mathbb{R}$ is the *covariance kernel*. We denote an implicitly-defined GP by $\mathcal{GP}(m, k)$.

We denoted the index set with $X$ and the random variable with $Y_{x_i}$, for $x_i \in X$ to highlight that a GP is a distribution over functions where $X$ is the input space and $Y = Y_{x_i \in X}$ is the output space. Thus the sample space of each $Y_{x_i}$ is $Y$. Let $X = \mathbb{R}$ and $Y = \mathbb{R}$ be an input and output space respectively and $\mathcal{F}$ denote a space of functions, where $(f \in \mathcal{F}) : X \to Y$ denotes a possible function from $X$ to $Y$. We can place a GP prior over $\mathcal{F}$ by writing $\mathcal{F} \sim \mathcal{GP}(m, k)$. In this view, a sample from a GP is a sample from all the random variables $Y_x = f(x), \forall x \in X$.

The function space $\mathcal{F}$ over which a GP is defined is implicitly described by the chosen mean function and the covariance kernel; we have placed a *GP prior over* $\mathcal{F}$. Popular kernels include the squared exponential (this is a basic 'all-rounder' kernel) and the Matern class of kernels [6]. The squared exponential kernel is given by

$$k_{c,\lambda}^{sq}(x, x') = c \exp(\frac{-(x - x')^T(x - x')}{\lambda^2}), \tag{1}$$

where $c \in \mathbb{R}^+$ and $\lambda \in \mathbb{R}$ are hyperparameters of the kernel. The parameter $c$ is the signal variance and the parameter $\lambda$ is the characteristic length scale[2]. The squared exponential kernel describes a function space of smooth functions that have a variability that is described by the signal variance and the length scale; the signal variable loosely specifies how much the function can vary from the mean function and the length scale specifies how quickly the function can vary from the mean function.

### 2.1 GP Prediction with Deterministic Inputs

Let $X = \mathbb{R}^d$ denote an input space, where $d$ denotes the dimension of $X$, and let $Y = \mathbb{R}$ denote an output space. Let $f : X \to Y$ be an arbitrary function from $X$ to $Y$ from a function space $\mathcal{F}$. We define a GP prior over $\mathcal{F}$ as $\mathcal{F} \sim \mathcal{GP}(m, k)$. The mean function $m$ and covariance kernel $k$ implicitly define the function space $\mathcal{F}$. Further, let the likelihood function be $\tilde{y} = f(x) + \epsilon$, where $\epsilon \in \mathcal{N}(0, \sigma_\epsilon^2)$ and $\tilde{y} \in (\tilde{Y} = Y)$ denotes a noisy observation. Finally, let $\mathbf{X} \in \mathbb{R}^{k \times d}$ be a matrix of $k$ input points from $X$, and let $\mathbf{Y} \in \mathbb{R}^{k \times 1}$ be a matrix of $k$ noisy output points from $\tilde{Y}$.

We are interested in the posterior GP over $\mathcal{F}$ given the data $(\mathbf{X}, \mathbf{Y})$. However, rather than finding an explicit conditional distribution over $\mathcal{F}$, we express the distribution of the posterior GP evaluated at a

---

[2]It is possible to write the squared exponential kernel in terms of a specific length scale for each dimension of the input space. For simplicity, will be using a single, common length scale for all dimensions in this article.

particular point $x^* \in X$, which, by Definition 2.1 is a Gaussian random variable. Therefore, for an arbitrary point $x^* \in X$, the predictive posterior density of the random variable $Y_{x^*} = f(x^*)$ is

$$Y_{x^*} = f(x^*) \sim p_{f(x^*)}(y^*|\mathbf{X}, \mathbf{Y}) = \mathcal{N}(\mu_*(x^*), \sigma_*^2(x^*)), \tag{2}$$

where,

$$\mu_* : X \to \mathbb{R}$$
$$x^* \mapsto \mu_*(x^*) = m(x^*) + k(x^*, \mathbf{X})(\mathbf{K} + \sigma_\epsilon^2 \mathbf{I})^{-1}(\mathbf{Y} - m(x^*)),$$

$$\tag{3}$$

$$\sigma_*^2 : X \to \mathbb{R}$$
$$x^* \mapsto \sigma_*^2(x^*) = k(x^*, x^*) - k(x^*, \mathbf{X})(\mathbf{K} + \sigma_\epsilon^2 \mathbf{I})^{-1} k(x^*, \mathbf{X})^T.$$

For a deterministic input, GP prediction can be thought of as a function from the input to a mean and a variance, as Equation 3 shows, that defines the predictive posterior distribution.

# 3 GP Prediction with Uncertain Inputs

A useful enhancement to GP prediction with deterministic input is to consider a distributional input. Let the input to the GP now be a random variable $X^*$, the sample space of which is $X$. Thus, $X^* \sim p_{X^*}(\cdot)$. We are still interested in the predictive distribution of $Y_{x^*} \sim p_{f(X^*)}(y|\mathbf{X}, \mathbf{Y})$, where $x^* \in X^*$. However, we must now marginalize over $X^*$ because it is a random variable:

$$p_{f(X^*)}(f(x^*) = y^*|\mathbf{X}, \mathbf{Y}) = \int_X p_{f(x^*)}(y^*|\mathbf{X}, \mathbf{Y}, x^*) p_X(x^*) \mathrm{d}x^*. \tag{4}$$

In general, this integral is intractable. Therefore, we must usually resort to approximation methods. We discuss two approximation methods used in the literature below. In Section 4, we introduce a method for implicitly computing the predictive posterior distribution with uncertain input using a state-of-the-art UTPA [1, 2].

## 3.1 Moment-matching approximation

We can approximate the intractable predictive posterior distribution as a Gaussian distribution. This is done by defining a Gaussian distribution using the first and second moments of the predictive posterior distribution [4, 5]. This mean and variance can be computed analytically when using the squared exponential kernel and the input is a Gaussian random variable [4, 5]. For more complicated input distributions and GP priors, it is difficult to find closed-form solutions. Furthermore, a Gaussian approximation to a non-Gaussian predictive posterior distribution would lack important statistical information such as multi-modality and skewness.

## 3.2 Monte Carlo simulation

The Monte Carlo method can be used to obtain a more informative approximation of $p_{f(X^*)}(y^*|\mathbf{X}, \mathbf{Y})$. Monte Carlo simulation [7, 8, 9] applies the Monte Carlo method to approximate the probability density function of a transformed random variable. That is, if we want the probability density function of $Y = f(X)$, then we first take $n$ samples of $x_i \sim X$, transform each sample by applying $f$ to obtain $n$ samples $y_i = f(x_i)$ of $Y$, and create a density estimation (such as a histogram) using the transformed samples. Better approximations are obtained by increasing the number of samples $n$.

Girard *et al.* [5] suggests estimating the integral in Equation 4 using Monte Carlo integration, where,

$$p_{f(X^*)}(y^*|\mathbf{X}, \mathbf{Y}) = \int p_{f(x^*)}(y^*|\mathbf{X}, \mathbf{Y}, x^*) p_X(x^*) \, \mathrm{d}x^* \approx \frac{1}{n} \sum_{i=1}^n p_{f(x^*)}(y^*|\mathbf{X}, \mathbf{Y}, x_i^*) \, \mathrm{d}x^*. \tag{5}$$

For this method, we would need to evaluate the sum in Equation 5 for each value that $f(x^*)$ can take. Instead, when using Monte Carlo simulation, we will obtain samples of $f(x^*)$ under $p_{f(x^*)}(y^*|\mathbf{X}, \mathbf{Y})$ by taking samples of $x_i^* \sim X^*$, calculating the mean $\mu_*(x_i^*)$ and variance $\sigma_*^2(x_i^*)$ according to Equation 3 for $p_{f(X^*)}(y^*|\mathbf{X}, \mathbf{Y}, x^*)$, and taking samples $y_i^* \sim \mathcal{N}(\mu_*(x_i^*), \sigma_*^2(x_i^*))$.

The Monte Carlo method can be used to obtain an arbitrarily-good approximation of $p_{f(X^*)}(y^*|\mathbf{X}, \mathbf{Y})$. However, doing so can be prohibitively expensive due to its slow convergence rate [9].

# 4 Gaussian Process Prediction with Uncertain Inputs on an Uncertainty-Tracking Processor Architecture

To overcome the accuracy limitations of moment-matching and the computational costs of Monte Carlo simulations, we present a method that is enabled by recent advances in UTPAs [1, 2]. A UTPA is a computer microarchitecture that can represent distributional information in its microarchitectural state and track how these distributions evolve under arithmetic operations transparently to the applications running on it. A UTPA does this by providing an in-processor representation of probability distributions and a suite of arithmetic operations that can manipulate this representation. A UTPA carries out *deterministic computations on probability distributions* and does not rely on any sampling methods. Therefore, unlike Monte Carlo methods that require convergence to a good solution by increasing the number of Monte Carlo iterations, a UTPA is *convergence-oblivious* up to the fidelity of its representation.

The UTPA presented by Tsoutsouras *et al.* [1, 2] has an associated *representation size*, $r$, that describes the *precision* of the representation. Larger representation sizes result in more accurate representations. A useful analogy is the IEEE-754 standard [10, 11] that approximately represents the infinite set of real numbers as floating point numbers on computers that can only handle a finite set of values.

A practical way of thinking of what a UTPA does is that it carries out uncertainty propagation. If we have a random variable $X$ represented in the UTPA's distribution representation and a function $f : X \to Y$, then a UTPA directly computes the distribution representation of $Y = f(X)$ by directly computing it through the function $f$ without resorting to Monte Carlo sampling.

## 4.1 Gaussian Process Prediction with Uncertain Inputs as a Transformation of Random Variables

Since a UTPA can represent any distribution for the input random variable $X$, our goal is to use a UTPA to directly compute the predictive posterior distribution of a GP with an uncertain input. To do this, we first note that a GP is practically two functions that map some input $x^*$ to the mean $\mu_*(x^*)$ and variance $\sigma_*^2(x^*)$ of the output distribution at $x^*$ (as given by Equation 3), but *not to the output random variable itself*. Simply plugging a UTPA-represented distribution to Equation 3 on a UTPA would result in distributions over the mean and variance and not a distribution of the output random variable that we desire.

We can express a transformation of random variables that results in a random variable that has the desired predictive posterior distribution $p_{f(X^*)}(y^*|\mathbf{X}, \mathbf{Y})$ by using the "reparametrization trick" [12],

$$f(X^*) = \mu_*(X^*) + \sigma_*(X^*)\epsilon, \tag{6}$$

where $\epsilon \sim \mathcal{N}(0, 1)$, and $\mu(x^*)$ and $\sigma_*(x^*)$ are the mean and standard deviation of the GP posterior at $x^*$ respectively from Equation 3. Theorem 4.1 shows that $f(X^*)$ has the desired distribution.

**Theorem 4.1.** Let $\mu_*(X^*)$ and $\sigma_*^2(X^*)$ denote the application of the mean and variance functions of the Gaussian Process predictive posterior distribution (as in Equation 3) to a random variable $X^* \sim p_X(\cdot)$ given some input and output data $\mathbf{X}$ and $\mathbf{Y}$ respectively, and let $\epsilon \sim \mathcal{N}(0, 1)$. Then, the probability density function of the random variable defined as

$$f(X^*|\mathbf{X}, \mathbf{Y}) = \mu_*(X^*) + \sigma_*(X^*)\epsilon$$

is given by

$$p_{f(X^*)}(y^*|\mathbf{X}, \mathbf{Y}) = \int p_{f(x^*)}(y^*|\mathbf{X}, \mathbf{Y}, x^*)p_X(x^*)\mathrm{d}x^* = \int \mathcal{N}\left(\mu_*(x^*), \sigma_*^2(x^*)\right) p_X(x^*)\mathrm{d}x^*,$$

where $p_{f(x^*)}(y^*|\mathbf{X}, \mathbf{Y}, x^*)$ is the conditional probability density function of $f(x^*)$ given $\mathbf{X}$, $\mathbf{Y}$, and a particular value of $x^* \in X^*$.

*Proof.* See Appendix B. ∎

## 4.2 Implementing the Transformation of Random Variables on an Uncertainty-Tracking Processor Architecture is Trivial

Using Algorithm 1, we can easily find the GP predictive posterior with an uncertain input using a UTPA. Here, $X^*$, $\epsilon$, and $y$ are distributional variables that are represented in the UTPA.

**Algorithm 1** Gaussian Process Prediction with Uncertain Inputs Using an Uncertainty-Tracking Processor Architecture [1, 2]

---

**Require:** $X^* \Longleftarrow \text{DistributionalVariable}, \mathbf{X}, \mathbf{Y}$
    $m \leftarrow \mu_*(X^*)$                                        ▷ From Equation 3
    $v \leftarrow \sigma_*^2(X^*)$                                        ▷ From Equation 3
    $\epsilon \leftarrow \text{GaussianDistributionalVariable}(0, 1)$      ▷ A distributional variable in the UTPA [1, 2]
    $y \leftarrow m + \sqrt{v}\epsilon$                                            ▷ From Equation 6
    **return** y

## 5 Methods

We compare the performance of Monte Carlo simulation and Algorithm 1 for computing the predictive posterior distribution. We measure the timing performance in terms of the wall-clock run time of each method and the accuracy in terms of the Wasserstein distance [13] of the result of each method compared to a ground-truth predictive posterior distribution obtained by a large ($n = 1,000,000$) Monte Carlo simulation.

We carry out experiments using the two methods across different number of iterations $n$ (of the Monte Carlo simulation) and representation sizes $r$ (of the UTPA) to compare the trade-offs between converging to the output distribution and the extra time required to do so. For each configuration of $n$ or $r$, we carry out 30 repetitions to account for variation in sampling[3].

We set the mean function of the GP prior to $m(x) = 0$. We set $X$ and $Y$ to be the real space $\mathbb{R}$, and generate a dataset $(\mathbf{X}, \mathbf{Y})$ of $k = 10$ data points by setting $x_0 = -3\pi$, $x_i = x_{i-1} + 0.6\pi$ [4], and $y_i$ as,

$$y_i(x_i) = \sin(2x_i)\cos(x_i)\left(e^{-\frac{1}{5}x} + e^{\frac{1}{5}x}\right) + \epsilon_i, \tag{7}$$

where each $\epsilon_i \sim \mathcal{N}(0, 0.01)$[5]. For each case, we use the squared exponential kernel $k^{sq}_{c=1,\lambda=\sqrt{1.7}}$ and we set the uncertain input distribution to $X^* \sim \mathcal{N}(0, 4)$ (see Figure 2 in Appendix C).

For the Monte Carlo method, we evaluate on $n \in \{4, 256, 1152, 2048, 4096, 8192, 16000, 32000, 64000, 128000, 512000\}$. For Algorithm 1, we evaluate on $r \in \{32, 64, 128, 256, 512, 2048\}$. For the UTPA for Algorithm 1, we use a commercial implementation of a multi-dimensional variation of the UTPA presented by Tsoutsouras *et al.* [1, 2] that automatically tracks correlations *between arbitrary random variables*.

We implement both Algorithm 1 and the Monte Carlo simulation using the C programming language and a custom tensor library implemented in C[6]. The commercial implementation of the UTPA [3] is implemented as a single-threaded emulated processor running on a 3rd Generation Intel Xeon system. As a result, Algorithm 1 is at an inherent disadvantage since it is benefiting only from the algorithmic advances of the distribution representation and its associated arithmetic and is not taking advantage of hardware acceleration; an FPGA or a custom silicon implementation of the underlying UTPA could provide even greater speedups than we present in this paper. We perform the Monte Carlo experiments on the same hardware as the hardware we used to emulate the UTPA. We note that we did not exploit parallelization for the Monte Carlo method (e.g., by using a GPU) since the commercial implementation of the UTPA [3] does not exploit parallelization either. See Appendix C for more detail on our method.

## 6 Results

Figure 1 summarizes our results; Table 2 in the Appendix shows numerical results for the other configurations. Figure 1a shows that the results from Algorithm 1 are almost always at the Pareto

---

[3]We calculate the Wasserstein distance for the result of Algorithm 1 by generating 1,000,000 samples from the UTPA representation. Therefore we repeat the experiments of Algorithm 1 on a UTPA 30 times for each representation size despite Algorithm 1 being deterministic and convergence-oblivious.

[4]In other words, we have equally divided the range $[-3\pi, 3\pi]$ to $n = 10$ points.

[5]We set the random seed to $42$.

[6]Code is available on GitHub: `https://github.com/physical-computation/uncertain-gaussian-process-code`

Table 1: Selection of results showing the Wasserstein distance and the run time required by the best overall configuration for Algorithm 1 and the close-to-equivalent Monte Carlo configuration. To obtain a similar accuracy, the Monte Carlo method took $\sim 108.80\times$ more time than Algorithm 1. See Table 2 for complete results.

| Problem | Core | Representation Size / Number of samples | Wasserstein Distance (mean $\pm$ std. dev.) | Run time (ms) (mean $\pm$ std. dev.) |
|---|---|---|---|---|
| Squared Exponential Kernel | Algorithm 1 | 128 | $0.00290 \pm 0.00050$ | $3.182 \pm 0.376$ |
| Squared Exponential Kernel | Traditional Monte Carlo | 128000 | $0.00324 \pm 0.00129$ | $346.186 \pm 1.800$ |

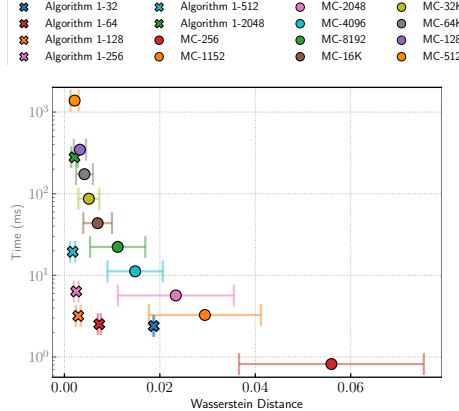

(a) Pareto plot summarizing our results. The Monte Carlo simulation with $n = 4$ is omitted for clarity.

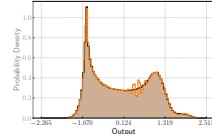

(b) Histogram for a single experiment using Algorithm 1 with $r = 128$. The Wasserstein distance was 0.00337307; the run time was 5.177 ms (67.3x faster).

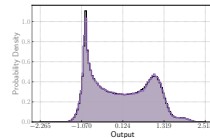

(c) Histogram for a single experiment using Monte Carlo with $n = 128000$. The Wasserstein distance was 0.00224596; the run time was 348.452 ms (67.3x slower).

Figure 1: Summaries of our key results. Subfigure (a) is a Pareto plot between the mean run time and the mean Wasserstein distance from the ground-truth output distribution. The error bars are $\pm 1$ standard deviation. From our experiments, Algorithm 1 is almost always on the Pareto frontier, even after accounting for uncertainty. Subfigures (b) and (c) are histograms from Algorithm 1 and the Monte Carlo method respectively. The ground-truth distribution is in black, behind each histogram.

frontier. Therefore, for any required accuracy except for the highest, in the ranges that we had tested, Algorithm 1 is at least an order of magnitude faster. Table 1 shows that for $r = 128$, the Monte Carlo method with $n = 128000$ shows a nearly identical Wasserstein distance (albeit with more than twice the standard deviation) while taking approximately $108.80\times$ longer to compute.

Furthermore, Algorithm 1 does not suffer from large variances in the accuracy since the UTPA carries out *deterministic computation on probability distributions*. We see this from the constant variance shown by the results from Algorithm 1. This uncertainty is due to the fact that we calculated the Wasserstein distance using finite samples from the output distribution. It is possible to calculate the Wasserstein distance directly from the UTPA's representation, but we did not do so at this time. After $r = 128$, the accuracy of Algorithm 1 stagnates and the trade-off with run time increases. Therefore, for this particular application, if larger accuracy is required, the Monte Carlo method must be used.

Figures 1b and 1c show that the posterior distribution is more complex than a Gaussian. A moment-matched Gaussian solution would, for example, miss that the posterior distribution is multi-modal.

## 7 Conclusions

Existing methods for approximating the often-intractable GP predictive posterior distribution with uncertain inputs can be inaccurate and constrained in their use (moment-matching), or computationally expensive (Monte Carlo simulation). We present a method for computing the desired predictive posterior distribution on a UTPA [1, 2] by making use of the reparametrization trick. Our method can be used on *any input distribution and GP prior* unlike moment-matching and we show experimentally that our method can be *as accurate as and orders of magnitude faster than Monte Carlo simulation*.

## 8   Acknowledgements

This work was supported by UKRI Materials Made Smarter Research Centre (EPSRC grant EP/V061798/1).

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

## Supplementary Section

## A Mathematical Preliminaries

In this section, we define some key mathematical details.

### A.1 Random Variables

**Definition A.1** (Probability density function)**.** Let $X$ be a set called the sample space and $p_X$ be a map from $X$ to $\mathbb{R}^+$,

$$p_X : X \to \mathbb{R}^+,$$

that satisfies:

$$\int_X p_X(x) \, \mathrm{d}x = 1.$$

We define $p_X$ to be the *probability density function* on $X$.

**Definition A.2** (Random Variable)**.** Let $X$ be a set and $p_X$ a probability density function on $X$. We define the tuple $(X, p_X)$ as a random variable[7].

In the rest of this article, we suppress the notation $(X, p_X)$ and simply write $X$ to denote a random variable. We use the simple definition of a random variable above rather than the typical measure-theoretic definition to capture what we think of as a random variable in practice; we often think of a random variable in terms of its probability density function (for example, when we think of a Gaussian random variable, we often think of the bell curve drawn by the Gaussian probability density function).

Often, the set $X = \mathbb{R}^d$, where $\mathbb{R}^d$ is the $d$-dimensional real space. In the integration above (and below), we have suppressed the fact that it contains multiple integrals over each dimension of $X$.

The probability distribution of a random variable is a set function $\mathbb{P}_X : \Omega \to \mathbb{R}^+$ that tells us about the probability of the random variable taking on the values inside the given set, where $\Omega$ is a set of subsets of $X$. $\Omega$ must technically be a $\sigma$-algebra, which for our purposes, can be thought of as a set of sets each of which can be expressed as a countable union of disjoint sets that also belong to $\Omega$[8]. Since we are only considering random variables that contain probability density functions, we define the distribution of a random variable as:

**Definition A.3** (Distribution of a Random Variable)**.** Given a random variable $X$ with probability density function $p_X$, the distribution of $X$ is given by the set function

$$\mathbb{P}_X : \Omega \to \mathbb{R}^+$$
$$\omega = \bigcup_i U_i \mapsto \mathbb{P}_X(\omega) = \sum_i \int_{U_i} p_X(x) \, \mathrm{d}x, \tag{8}$$

where $U_i$ is a collection of disjoint sets whose union equals the input set $\omega$.

**Definition A.4** (Cumulative Distribution Function of a Random Variable)**.** The cumulative distribution function $F_X$ of the random variable $X$ with probability density function $p_X$ is given by

$$F_X : X \to [0, 1]$$
$$x \mapsto F_X(x) = \mathbb{P}(\{t | t \in X \text{ and } t < x\}) = \int_{\{t | t \in X \text{ and } t < x\}} p_X(t) \, \mathrm{d}t, \tag{9}$$

Note that in the integrations above, we have suppressed the fact that the integrations need to occur over dimentions of $X$. By applying the Fundamental Theorem of Calculus to each such integral, we obtain that $p_X(x) = \frac{\partial^d F_X}{\partial x_1 \dots \partial x_d}(x)$ This also gives us a way of going from a distribution of a random

---

[7]There can be measure theoretic random variables, such as the random variable that has the Cantor distribution, that do not admit probability density functions [14]. In this article, we do not consider such exotic random variables.

[8]The sets in a $\sigma$-algebra must also satisfy that countable unions and intersections also belong to $\Omega$.

variable to its probability density function. Therefore, in this article, we often refer to the distribution and the probability density function interchangeably.

An example of a high dimensional random variable is the multivariate Gaussian random variable:

**Definition A.5** (Multivariate Gaussian random variable). Let $\boldsymbol{\mu} \in \mathbb{R}^d$ and $\boldsymbol{\Sigma} \in \mathbb{R}^{d \times d}$ be a mean vector and a covariance matrix respectively. The multivariate Gaussian random variable $(X^{\mathcal{N}}, p_X^{\mathcal{N}})$ is defined as:

$$X^{\mathcal{N}} = \mathbb{R}^d,$$

$$p_X^{\mathcal{N}} : \mathbb{R}^d \to \mathbb{R}^+,$$

$$x \mapsto \frac{1}{\sqrt{(2\pi)^d \det \boldsymbol{\Sigma}}} \exp\left(-\frac{1}{2}(x - \boldsymbol{\mu})^T \boldsymbol{\Sigma}^{-1}(x - \boldsymbol{\mu})\right).$$

$p_X^{\mathcal{N}}$ is called the multivariate Gaussian distribution. A Gaussian distribution is fully specifed by its mean vector $\mu$ and its covariance matrix $\boldsymbol{\Sigma}$. Therefore, we can also write that a random variable $X^{\mathcal{N}}$ is distributed as a multivariate Gaussian by writing

$$X^{\mathcal{N}} \sim \mathcal{N}(\boldsymbol{\mu}, \boldsymbol{\Sigma}).$$

The mean vector $\boldsymbol{\mu}$ is the expectation of the random variable $X^{\mathcal{N}}$, which is defined for any random variable $X$ as,

**Definition A.6** (Expectation of a Random Variable). Let $(X, p_X)$ be a random variable. The expectation of $X$ is defined as:

$$\mathbb{E}[X] = \int_X x p_X(x) \, \mathrm{d}x.$$

An alternative view of a multivariate Gaussian random variable, or any other random variable over $\mathbb{R}^d$, is that it is a collection of univariate (single dimensional) random variables that are jointly distributed as the multivariate distribution. The covariance matrix $\boldsymbol{\Sigma}$ is a matrix the elements $\Sigma_{ij} = \mathrm{cov}[X_i, X_j]$, where $X_i$ and $X_j$ are the $i$-th and $j$-th univariate random variables of $X$ respectively. Generally, the covariance of two random variables is defined as:

**Definition A.7** (Covariance of Random Variables). Let $(X, p_X)$ and $(Y, p_Y)$ be two random variables. The covariance of $X$ and $Y$ is defined as:

$$\mathrm{cov}[X, Y] = \mathbb{E}[(X - \mathbb{E}[X])(Y - \mathbb{E}[Y])].$$

We note that the covariance between the same random variable is its variance:

$$\mathrm{cov}[X, X] = \mathbb{E}[(X - \mathbb{E}[X])^2] = \mathrm{var}[X].$$

In this article, we write $\mathrm{cov}[X]$ to denote the variance of $X$.

Give a collection of random variables $\{X_i\}_{i=1,\ldots,n}$, we can construct a covariance matrix $\boldsymbol{\Sigma}$:

**Definition A.8** (Covariance Matrix). Let $X$ be an index set and $\{Y_i\}x = 1^n$ be a collection of random variables indexed by $x \in X$. Then, the covariance matrix $\boldsymbol{\Sigma} \in \mathbb{R}^{n \times n}$ of the collection of random variables that contains the pairwise covariances of the random variables:

$$\Sigma_{ij} = \mathrm{cov}[Y_i, Y_j].$$

## A.2 Change of variables

Let $f : X \to Y$ denote a *transformation* from a random variable $X$ to a random variable $Y$ (i.e., $Y = f(X)$)[9]. We can also apply $f$ to an instance value $x$ of $X$ to obtain an instance value $y = f(x)$ of $Y$.

If $f$ is invertible and once-differentiable, then Theorem A.1 derives the probability density function of $Y$, denoted as $p_Y$ [15, Chapter 3.7].

---

[9]$f$ transforms the set $X$ such that there exists a valid probability density function $p_Y$ over the set $Y$.

**Theorem A.1** (Change of variables). Given a random variable $X$ with a probability density function $p_X$ and an invertible and once-differentiable transformation $f : X \to Y$, the probability density function $p_Y$ of the random variable $Y = f(X)$ is given by:

$$p_Y : Y \to \mathbb{R}^+,$$
$$y \mapsto p_Y(y) = p_X \circ f^{-1}(y) |\det \nabla f\left(f^{-1}(y)\right)|^{-1}$$
$$= p_X \circ f^{-1}(y) |\det \nabla f^{-1}(y)|,$$

where $f^{-1}$ is the inverse of $f$ and $\nabla f(\,\cdot\,)$ and $\nabla f^{-1}(\,\cdot\,)$ denote the Jacobian matrices of $f$ and $f^{-1}$ respectively.

## A.3 Gram matrices, kernels and covariance functions

**Definition A.9** (Kernel). Let $X$ be an input space. A kernel on $X$ is a function $k : X \times X \to \mathbb{R}$.

In this paper, $X$ will always be a subset of $\mathbb{R}^d$. Given a kernel $k$ on $X$, we can construct a Gram matrix:

**Definition A.10** (Gram matrix). Let $\{x_i \in X\}_{i=1,\ldots,n}$ be a set of $n \in \mathbb{N}$ points in $X$, and $k$ be a kernel on $X$. The Gram matrix $\mathbf{K} \in \mathbb{R}^{n \times n}$ is a $n \times n$ matrix, where each element $K_{ij} = k(x_i, x_j)$.

Kernels can have some properties:

**Definition A.11** (Symmetric kernel). A kernel $k$ on $X$ is said to be symmetric if $k(x, x') = k(x', x)$ for all $x, x' \in X$.

**Definition A.12** (Postive semi-definite kernel). A kernel $k$ on $X$ is said to be positive semi-definite if it satisfies:

$$\int_X \int_X k(x, x') f(x) f(x') \mathrm{d}x \mathrm{d}x' \geq 0,$$

for all square-integrable $f : X \to \mathbb{R}$.

An alternative definition can be written in terms of the kernel's Gram matrix. A Gram matrix $\mathbf{K}$ is positive semi-definite if $\mathbf{v}^T \mathbf{K} \mathbf{v} \geq 0$ for all $\mathbf{v} \in \mathbb{R}^n$. If, $\forall n \in \mathbb{N}$ and all sets of $n$ points $\{x_i \in X\}_{i=1,\ldots,n}$ the Gram matrix $\mathbf{K}$ is positive semi-definite, then the kernel $k$ is defined to be positive semi-definite.

A kernel $k$ on $X$ which is symmetric and positive semi-definite is called a covariance function because it can be used to define a valid covariance matrix.

**Definition A.13** (Covariance function). A covariance function $k$ on $X$ is a kernel $k : X \times X \to \mathbb{R}$ which is symmetric and positive semi-definite. A covariance function is also called a covariance kernel.

The Gram matrix of a covariance function is also symmetric and positive semi-definite, which means that it can act as a covariance matrix.

### A.3.1 Some notes on notation

In this article, we do not make the distinction between a scalar and a vector. Rather, we write $x \in (X = \mathbb{R}^d)$ to denote an element of some $d-$dimensional real space, where $d$ should be clear from context, or is an abstract value. Thus, $x$ contains $d$ elements, and when $d = 1$, $x$ represents a scalar. Any vector and matrix operation we write also works for scalar values.

Furthermore, we will be applying a kernel function $k$ to a variety of different types of inputs, with slight abuse of notation. Let us make clear what we mean by each kind here:

**Two vector inputs:** $k(x, x')$ is the kernel applied to two points $x, x' \in X$. Thus the output is a scalar value $k(x, x') \in \mathbb{R}$.

**A vector input and a matrix input:** $k(x, \mathbf{X})$ is the kernel applied to a point $x \in X$ and a set of points $n$ points $\{x_i\}_{i=1,\ldots,n}$ collected into a matrix $\mathbf{X} \in \mathbb{R}^{n \times d}$. The output is a vector $k(x, \mathbf{X}) \in \mathbb{R}^n$, where each element $k(x, \mathbf{X})_i = k(x, x_i)$.

**Two matrix inputs:** $k(\mathbf{X}, \mathbf{X}')$ is the kernel applied to two sets of matrices obtained from sets $\{x_i\}_{i=1,\ldots,n}$ and $\{x'_j\}_{i=1,\ldots,m}$ collected into matrices $\mathbf{X} \in \mathbb{R}^{n \times d}$ and $\mathbf{X}' \in \mathbb{R}^{m \times d}$ respectively. The output is a matrix $k(\mathbf{X}, \mathbf{X}') \in \mathbb{R}^{n \times m}$, where each element $k(\mathbf{X}, \mathbf{X}')_{ij} = k(x_i, x'_j)$.

# B  Proof of Theorem 4.1

**Theorem B.1.** Let $\mu_*(X^*)$ and $\sigma_*^2(X^*)$ denote the application of the mean and variance functions of the Gaussian Process predictive posterior distribution (Equation 3) to a random variable $X^* \sim p_X(\cdot)$ given some input and output data $\mathbf{X}$ and $\mathbf{Y}$ respectively, and let $\epsilon \sim \mathcal{N}(0, 1)$. Then, the probability density function of the random variable defined as

$$f(X^*|\mathbf{X}, \mathbf{Y}) = \mu_*(X^*) + \sigma_*(X^*)\epsilon$$

is given by

$$p_{f(X^*)}(y^*|\mathbf{X}, \mathbf{Y}) = \int p_{f(x^*)}(y^*|\mathbf{X}, \mathbf{Y}, x^*)p_X(x^*)\mathrm{d}x^* = \int \mathcal{N}\left(\mu_*(x^*), \sigma_*^2(x^*)\right) p_X(x^*)\mathrm{d}x^*,$$

where $p_{f(x^*)}(y^*|\mathbf{X}, \mathbf{Y}, x^*)$ is the conditional probability density function of $f(x^*)$ given $\mathbf{X}$, $\mathbf{Y}$, and a particular value of $x^* \in X^*$.

*Proof.* The conditional probability density function of $f(x^*)$ given $\mathbf{X}$, $\mathbf{Y}$, and a particular value of $x^* \in X^*$ can be derived from Theorem A.1 for the transformation in the theorem:

$$p_{f(x^*)}(y^*|x^*, \mathbf{X}, \mathbf{Y}) = p_\epsilon\left(\frac{y^* - \mu_*(x^*)}{\sigma_*^2(x^*)}\right) \frac{\mathrm{d}}{\mathrm{d}y^*}\left(\frac{y^* - \mu_*(x^*)}{\sigma_*^2(x^*)}\right)$$

$$= \frac{1}{\sqrt{2\pi}} \exp\left(-\frac{1}{2}\left(\frac{y^* - \mu_*(x^*)}{\sigma_*^2(x^*)}\right)^2\right) \frac{1}{\sigma_*^2(x^*)}$$

$$= \mathcal{N}\left(\mu_*(x^*), \sigma_*^2(x^*)\right).$$

Then, the joint probability density function $p_{f(x^*)}(f(x^*), x^*)$ is given by

$$p_{f(x^*)}(y^*, x^*|\mathbf{X}, \mathbf{Y}) = p_{f(x^*)}(y^*|x^*, \mathbf{X}, \mathbf{Y})p_X(x^*),$$

and the marginal probability density function of $f(x^*)$ is given by:

$$p_{f(X^*)}(y^*|\mathbf{X}, \mathbf{Y}) = \int p_{f(x^*)}(y^*|x^*, \mathbf{X}, \mathbf{Y})p_X(x^*)\mathrm{d}x^*$$

$$= \int \mathcal{N}\left(\mu_*(x^*), \sigma_*^2(x^*)\right) p_X(x^*)\mathrm{d}x^*.$$

∎

# C  Methods: More details

## C.1  The Gaussian Process and Input Distribution used in Experiments

We set the prior mean function to $m(x) = 0$. We set $X = \mathbb{R}$ and $Y = \mathbb{R}$, and generate a dataset $(\mathbf{X}, \mathbf{Y})$ of $k = 10$ data points by setting $x_0 = -3\pi$, $x_i = x_{i-1} + 0.6\pi$[10], and $y_i$ as

$$y_i(x_i) = \sin(2x_i)\cos(x_i)\left(e^{-\frac{1}{5}x} + e^{\frac{1}{5}x}\right) + \epsilon_i, \tag{10}$$

where each $\epsilon_i \sim \mathcal{N}(0, 0.1)$[11]. For each case, we use the squared exponential kernel $k^{sq}_{c=1, \lambda=\sqrt{1.7}}$ and we set the uncertain input distribution to $X^* \sim \mathcal{N}(0, 4)$.

Figure 2 show an illustration of the Gaussian Process and the input distribution used.

---

[10]In other words, we have equally divided the range $[-3\pi, 3\pi]$ to $n = 10$ points.

[11]We set the random seed to 42.

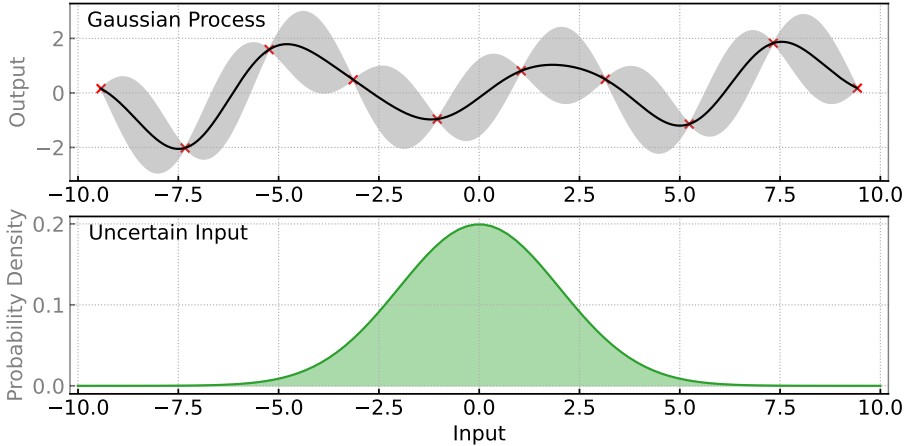

Figure 2: Illustration of the Gaussian Process (with its data points) and the input distribution that are used throughout the experiments in this article.

## C.2 Generating random samples for the Monte Carlo method

The Monte Carlo method requires us to generate samples from several Gaussian distributions. For this, we used the `gsl_ran_gaussian_ziggurat` function provided by GSL [16].

## C.3 Measuring the run time

We measure the run time as the sum of the time taken to generate samples incurred during the sampling step of the Monte Carlo method or the initializing step of Algorithm 1, and the time taken for the evaluation step. For both methods, we measure time using the `gettimeofday` function from the Standard C library [17]. We measure the time from the start of the main entry point until the end of key computations. The reported times omit any time spent by the programs on saving and reporting the results.

In order to explicitly quantify the *post-processing* step of the Monte Carlo method, we compute the mean and the variance of the samples obtained from the Monte-Carlo-based experiments. In Algorithm 1 we calculate the mean and variance using the representation of the commercial UTPA [3] as provided by its hardware API. We note that we are being generous to the Monte Carlo method, since the mean and the variance alone does not fully capture the shape of a non-Gaussian distribution. In contrast, the representation the uncertainty-tracking processor architecture (UTPA) captures the full distribution in its representation.

## C.4 Measuring the Wasserstein Distance

The Wasserstein distance [13] is a metric that measures the distance between probability distributions. We quantify the distance of the outputs to the ground-truth using the Wasserstein distance between the output distribution calculated by each approach and the ground-truth output distribution. We compute the ground-truth output distribution by running the Monte Carlo method with 1,000,000 samples. In our experiments, we calculate the Wasserstein distance using the `scipy.stats.wasserstein_distance` function from the `SciPy` Python package [18].

## C.5 Experimental setup

For Algorithm 1, we vary the representation size of the UTPA. Since the underlying UTPA on which Algorithm 1 is implemented generates in-processor representations of the output distribution, we take samples from this distributional representation to compute the Wasserstein distance. We take 1,000,000 samples, similar to the ground truth. We do not include the time taken for this sampling in the run time because this sampling is done solely to calculate the Wasserstein distance and is not part of a typical use case of Algorithm 1. The commercial implementation [3] of the UTPA

| Problem | Core | Representation Size / Number of samples | Wasserstein Distance (mean ± std. dev.) | Run time (ms) (mean ± std. dev.) |
|---|---|---|---|---|
| Squared Exponential Kernel | Algorithm 1 | 32 | $0.01871 \pm 0.00010$ | $2.392 \pm 0.359$ |
| Squared Exponential Kernel | Algorithm 1 | 64 | $0.00730 \pm 0.00031$ | $2.524 \pm 0.052$ |
| **Squared Exponential Kernel** | **Algorithm 1** | **128** | $\mathbf{0.00290 \pm 0.00050}$ | $\mathbf{3.182 \pm 0.376}$ |
| Squared Exponential Kernel | Algorithm 1 | 256 | $0.00246 \pm 0.00050$ | $6.348 \pm 0.148$ |
| Squared Exponential Kernel | Algorithm 1 | 512 | $0.00173 \pm 0.00053$ | $19.458 \pm 2.215$ |
| Squared Exponential Kernel | Algorithm 1 | 2048 | $0.00211 \pm 0.00069$ | $279.482 \pm 2.491$ |
| Squared Exponential Kernel | Traditional Monte Carlo | 4 | $0.41230 \pm 0.22109$ | $0.129 \pm 0.006$ |
| Squared Exponential Kernel | Traditional Monte Carlo | 256 | $0.05596 \pm 0.01936$ | $0.819 \pm 0.015$ |
| Squared Exponential Kernel | Traditional Monte Carlo | 1152 | $0.02943 \pm 0.01174$ | $3.268 \pm 0.087$ |
| Squared Exponential Kernel | Traditional Monte Carlo | 2048 | $0.02334 \pm 0.01216$ | $5.673 \pm 0.023$ |
| Squared Exponential Kernel | Traditional Monte Carlo | 4096 | $0.01482 \pm 0.00580$ | $11.195 \pm 0.051$ |
| Squared Exponential Kernel | Traditional Monte Carlo | 8192 | $0.01116 \pm 0.00578$ | $22.316 \pm 0.092$ |
| Squared Exponential Kernel | Traditional Monte Carlo | 16000 | $0.00696 \pm 0.00301$ | $43.553 \pm 0.654$ |
| Squared Exponential Kernel | Traditional Monte Carlo | 32000 | $0.00512 \pm 0.00219$ | $86.781 \pm 0.447$ |
| Squared Exponential Kernel | Traditional Monte Carlo | 64000 | $0.00419 \pm 0.00177$ | $173.586 \pm 1.463$ |
| **Squared Exponential Kernel** | **Traditional Monte Carlo** | **128000** | $\mathbf{0.00324 \pm 0.00129}$ | $\mathbf{346.186 \pm 1.800}$ |
| Squared Exponential Kernel | Traditional Monte Carlo | 512000 | $0.00213 \pm 0.00085$ | $1385.516 \pm 8.283$ |

Table 2: Complete results comparing the mean Wasserstein distance (to the ground truth) obtained by the Monte Carlo method and Algorithm 1 to the mean run time taken to run each configuration. In order to obtain a similar accuracy, the Monte Carlo method took approximately $108.80\times$ more time than Algorithm 1. In order for the Monte Carlo method to obtain better results than Algorithm 1 (at $r = 128$) by (approximately) more than 1-standard deviation, the Monte Carlo must spend $435.42\times$ more time.

presented by Tsoutsouras *et al.* [1, 2] is implemented as a single-threaded emulated processor running on top of a single x86 3rd Generation Intel Xeon system. As a result, our implementation is at an inherent disadvantage since it is benefiting only from the algorithmic advances of the distribution representation and its associated arithmetic and is not taking advantage of hardware acceleration; an FPGA or custom silicon implementation could provide even greater speedups than we present in this paper.

For the Monte Carlo simulations, we vary the number of Monte Carlo iterations that were used in each simulation. We perform the Monte Carlo experiments on the same x86 hardware as the hardware we used to emulate the UTPA. This provides a baseline for the performance of the Monte Carlo method that can be expected in the real-world. All random samples for the Monte Carlo simulation were carried out using functions made available by GSL [16].

We implement both Algorithm 1 and Monte Carlo experiments using the C programming language and a custom tensor library implemented in C. We compile Monte Carlo implementations with the highest level of compiler optimizations (`-O3`)[12].

# D    Ensuring meaningful timing results

When running the experiments on the Monte Carlo method, each repetition of an experiment was run after a 5s delay. This delay ensures that we avoid buffer cache optimizations carried out by the operating system.

We note that we did not exploit parallelization for the Monte Carlo method (e.g., by using a GPU) since the commercial implementation of the UTPA [3] does not exploit parallelization either. We felt that this provided an apples-to-apples comparison.

# E    Extended Table of Results

Table 2 shows the numerical results from all our experiments.

---

[12]Code is available on GitHub: `https://github.com/physical-computation/uncertain-gaussian-process-code`

