# OpenReview forum: "Gaussian Process Predictions with Uncertain Inputs Enabled by Uncertainty-Tracking Processor Architectures"
_NeurIPS.cc/2024/Workshop/MLNCP — MLNCP Poster_

### Official Review · Reviewer_bfFF · 2024-10-03
**Review for Gaussian Process Predictions with Uncertain Inputs Enabled by Uncertainty-Tracking Microprocessors**

**Rating:** 7
**Confidence:** 3

**Review:**

The paper discusses how uncertainty-tracking microprocessors can be used to efficiently compute posterior distributions with a Gaussian process prior applied to uncertain inputs. Typical previous approaches using Gaussian process priors required the input to be deterministic. For uncertain inputs, some approaches only approximated the posterior with moment-matching for the first few moments, but this often cannot represent many statistical properties of posterior distributions including additional modes. Alternatively, computationally-expensive Monte Carlo simulation can provide more accurate estimates of the posterior. In this work, the authors proposed to use a microarchitecture inherently suited for modelling distributions to make accurate estimates of the posterior distribution for uncertain inputs computationally tractable.

Before the methods section, the paper provides a detailed and very helpful derivation of the previous approaches for computing posterior distributions with Gaussian processes for deterministic and stochastic outputs. This section is well-written and effectively defines the vocabulary and related work in the field. It also clearly highlights the challenges with the existing moment-matching and Monte Carlo estimation methods.

Comparatively, the discussion on the Laplace microprocessor is lacking in details. While the cited papers do discuss how the microprocessor implements the distribution-to-distribution operations, it would be good to have at least a couple examples of arithmetic operations and distribution representations in this paper to provide a bit more specificity to the discussion on how the Laplace microprocessor works. As is, the current discussion on the Laplace processor seems a bit too vague.

It would also be useful to provide more detail on why the hardware configuration for the Monte Carlo method (Apple M1 Pro) is a suitable comparison for the Laplace microprocessor. Would a desktop GPU be a better estimate of a hardware that is well suited for Monte-Carlo simulation in a way that is similar to how the Laplace microprocessor is well-suited for the proposed approach? Providing timing for the Monte Carlo approach across a variety of hardware configurations would be another way to mitigate this concern.

Overall, this paper provides a meaningful contribution by showcasing how uncertainty-tracking microprocessors could be very useful for communities interested in Gaussian Process-based uncertainty estimation.

---

### Official Review · Reviewer_yRUy · 2024-10-04
**Review of Paper Submission 50**

**Rating:** 7
**Confidence:** 4

**Review:**

This paper describes an application of uncertainty-tracking microprocessor on gaussian process predication.
It shows its advantages of high prediction accuracy and faster runtime against Monte Carlo simulation.
The writing is clear and easy to follow. Here is a list of pros and cons:

**Pros:**
* This paper identifies the limitation and challenges of exisiting methods of uncertain gaussian process predication.
* It theoretically demonstrates how this problem can be addressed by uncertainty-tracking microprocessors (Laplace) with proof.
* The results shows strong improvements in terms of runtime and prediction quality against Monte Carlo simulation, which implies its promising applications in practice.

**Cons:**
* The application of Laplace to solve uncertain gaussian process prediction needs more details to explain how it works. The current Section 4 only highlights the unique properties of Laplace to convey the high-level idea. As a reader, I have to go through the paper of reference [3] to have a much better understanding.

* The implmentation setup and runtime measurement details of the experiment results needs more clarification. In Appendix, this paper mentions that it uses Apple M1 Pro as the processor to run Mont Carlo simulation without parallel execution.
    * Which type of M1-Pro's core is used? Efficient core or Performance core?
    * Is M1-Pro run with battery mode or power plug-in mode?
    * What software language is used to implement Mont-Carlo simulation? C/C++, Python, or Matlab? (A reference to [9] is not helpful at all)

* The details of Laplace hardware is not clear. The referenced website [14] is not telling what kind of implementation of this microprocessor is used. Is it an emulated processor in RISC-V CPUs, x86 CPUs or FPGAs?

* It will be more insightful to shows the comparison results on some real-world uncertain Gaussian process predication problems. The current problem is too small to stress any processor cache/memory sub-system bound. In particular, the paper will be stronger to show results on larger problems that need to store long chain of ancestor addresses.

---

### Decision · Program_Chairs · 2024-10-10

Accept (Poster)